

# Drought stress strengthens the link between chlorophyll fluorescence parameters and photosynthetic traits

Jie Zhuang[1,*], Yonglin Wang[1,*], Yonggang Chi[1,2], Lei Zhou[1,3], Jijing Chen[1], Wen Zhou[1], Jun Song[1], Ning Zhao[1] and Jianxi Ding[1]

[1] College of Geography and Environmental Sciences, Zhejiang Normal University, Jinhua, Zhejiang, China
[2] State Key Laboratory of Resources and Environmental Information System, Institute of Geographic Sciences and Natural Resources Research, Chinese Academy of Sciences, Beijing, China
[3] Key Laboratory of Ecosystem Network Observation and Modeling, Institute of Geographic Sciences and Natural Resources Research, Chinese Academy of Sciences, Beijing, China
[*] These authors contributed equally to this work.

## ABSTRACT

Chlorophyll fluorescence (ChlF) has been used to understand photosynthesis and its response to climate change, particularly with satellite-based data. However, it remains unclear how the ChlF ratio and photosynthesis are linked at the leaf level under drought stress. Here, we examined the link between ChlF ratio and photosynthesis at the leaf level by measuring photosynthetic traits, such as net $CO_2$ assimilation rate ($A_n$), the maximum carboxylation rate of Rubisco ($V_{cmax}$), the maximum rate of electron transport ($J_{max}$), stomatal conductance ($g_s$) and total chlorophyll content ($Chl_t$). The ChlF ratio of the leaf level such as maximum quantum efficiency of PSII ($F_v/F_m$) is based on fluorescence kinetics. ChlF intensity ratio ($LD_{685}/LD_{740}$) based on spectrum analysis was obtained. We found that a combination of the stomatal limitation, non-stomatal limitation, and $Chl_t$ regulated leaf photosynthesis under drought stress, while $J_{max}$ and $Chl_t$ governed the ChlF ratio. A significant link between the ChlF ratio and $A_n$ was found under drought stress while no significant correlation in the control, which indicated that drought stress strengthens the link between the ChlF ratio and photosynthetic traits. These results suggest that the ChlF ratio can be a powerful tool to track photosynthetic traits of terrestrial ecosystems under drought stress.

# INTRODUCTION

The duration and frequency of drought are expected to increase due to global warming (*Pachauri et al., 2014*). Drought stress increases the frequency of forest fires and the death rate of trees (*Anderegg, Kane & Anderegg, 2013*; *Phillips et al., 2009*), limits leaf photosynthesis and plant productivity (*Akhkha, Boutraa & Alhejely, 2011*; *Tezara et al., 2003*) and decreases the gross primary production (GPP) (*Lee et al., 2013*; *Li et al., 2019*). The terrestrial ecosystem GPP driven by leaf photosynthesis is tightly related to chlorophyll fluorescence (ChlF) (*Murchie & Lawson, 2013*). In the light reaction of leaf photosynthesis,

Corresponding authors
Yonggang Chi,
chiyongg@mails.ucas.ac.cn
Lei Zhou, zhoulei@zjnu.cn

one of the main de-excitation processes for light absorption of the light-harvesting pigments is the emission of ChlF (*Aasen et al., 2019*). At the regional scale, solar-induced chlorophyll fluorescence (SIF) is observed based on solar irradiance and vegetation irradiance (*Smith et al., 2018*). ChlF opens a new perspective as a functional proxy of the terrestrial ecosystem GPP (*He et al., 2019*). However, it remains uncertain whether the link between ChlF and photosynthetic traits will be constrained by drought stress.

The net $CO_2$ assimilation rate ($A_n$) of leaves decreases during drought stress due to stomatal limitation and non-stomatal limitation (*Ashraf & Harris, 2013*; *Chaves et al., 2002*; *Flexas & Medrano, 2002*). On the one hand, stomatal limitation means that drought stress affects the diffusion process of $CO_2$ from the stomata to the intercellular spaces, and then reduces $A_n$ (*Flexas et al., 2014*; *Lawlor & Tezara, 2009*). A study of pine seedlings found that the limitation of stomatal conductance ($g_s$) to $A_n$ increased during drought stress (*Anev et al., 2016*). On the other hand, non-stomata limitations include biochemical limitation and mesophyll conductance ($g_m$) limitation (*Salmon et al., 2020*). Drought stress decreases the maximum carboxylation rate of Rubisco ($V_{cmax}$) and the maximum rate of electron transport ($J_{max}$) then reduces $A_n$ (*Flexas et al., 2004*; *Niinemets & Keenan, 2014*; *Rho et al., 2012*). A study of *Eucalyptus* and *Quercus* found that the $V_{cmax}$ and $J_{max}$ in drought were significantly lower than those in the control (*Zhou et al., 2014*). Also, drought stress leads to a reduction of $g_m$, which limits the diffusion of $CO_2$ from the leaf intercellular spaces to the sites of the dark reactions of photosynthesis in chloroplasts (*Flexas et al., 2012*; *Rancourt, Ethier & Pepin, 2015*). Recent studies have incorporated the effects of stomatal and non-stomatal limitations for predicting the response of photosynthesis to drought stress (*Drake et al., 2017*; *Salmon et al., 2020*). Thus, understanding the relative contributions of stomatal limitation versus non-stomatal limitation to the decline of $A_n$ is fundamental to project the effect of drought stress (*Campos et al., 2014*; *Chen, Yu & Huang, 2015*; *Gimeno et al., 2019*).

ChlF is a fast, accurate, and non-destructive probe which can be utilized to obtain information about the metabolism of photosystem II (PSII) (*Baker, 2008*). Photosynthetically active radiation is absorbed by chlorophyll and accessory pigments of chlorophyll-protein complexes and migrated to the reaction centers of photosystems I (PSI) and II (PSII), where the conversion of the quantum photosynthetic process takes place, and is then consumed by photochemistry, heat dissipation, or re-emitted as ChlF (*Porcar-Castell et al., 2014*). Due to the competition between these three processes, ChlF can be used to obtain photosynthesis information (*Maxwell & Johnson, 2000*; *Murchie & Lawson, 2013*). In recent years, researchers have used the changes of ChlF to explore photosynthetic apparatus under different environmental situations (*Badr & Brueggemann, 2020*; *Hajihashemi et al., 2020*; *Iqbal et al., 2019*; *Xu et al., 2020*).

Most studies on ChlF are based on polyphasic fluorescence transient (OJIP) to obtain fluorescence kinetic parameters, such as the maximum quantum efficiency of PSII reaction centers ($F_v/F_m$), the photochemical (qP) and non-photochemical (NPQ) quenching (*Mathobo, Marais & Steyn, 2017*). By analyzing the fluorescence kinetic curves, we can obtain abundant information about the structure and the function of PSII during stress conditions (*Krause & Weis, 1991*; *Stirbet et al., 2018*). $F_v/F_m$ is the maximal quantum

efficiency of PSII reaction centers which positively correlated with the activity of primary PSII photochemistry (*Butler, 1978*; *Stirbet et al., 2018*). Low $F_v/F_m$ represents that light energy absorbed by PSII reaction centers may be underutilized (*Fracheboud & Leipner, 2003*). PSII is considered to be a susceptible component of the photosynthetic machinery and will often bear the brunt of stress conditions, which leads to a decrease in $F_v/F_m$ (*Demmig-Adams & Adams, 2018*; *Long, Humphries & Falkowski, 1994*). For example, a study of *Viburnum* found that the $F_v/F_m$ significantly decreased during a severe drought (*Tribulato et al., 2019*). Likewise, *Li et al. (2008)* analyzed the effect of drought stress on the photochemical efficiency of leaves and found that the $F_v/F_m$ was decreased while the NPQ increased during severe drought stress. The decrease in $F_v/F_m$ indicates the down-regulation of photosynthesis or photoinhibition under stress (*Lichtenthaler & Rinderle, 1988*; *Van Kooten, 1990*). Therefore, fluorescence kinetic parameters have been used to determine photosynthetic traits successfully.

Laser-induced fluorescence spectrum analysis is a specific technique that provides a new approach to monitor vegetation physiology remotely (*Gouveia-Neto et al., 2011*; *Utkin et al., 2014*). Leaves have two fluorescence emission peaks located in the 685 nm of the red region ($LD_{685}$) and the 740 nm of the far-red region ($LD_{740}$) (*Buschmann, 2007*), which are closely related to the chlorophyll content ($Chl_t$) (*Kalmatskaya, Karavaev & Gunar, 2016*; *Nyachiro et al., 2001*). $LD_{685}$ and $LD_{740}$ both increase with the increases of $Chl_t$ at low $Chl_t$, while in the case of higher $Chl_t$, $LD_{685}$ will decrease due to re-absorption of the emitted red band fluorescence by the chlorophyll absorption bands (*Baker, 2008*; *Buschmann, 2007*). It has been demonstrated that $LD_{685}/LD_{740}$ is a good inverse indicator of the $Chl_t$ and reflects the active degree of photosynthesis (*Baker, 2008*; *D'Ambrosio, Szabo & Lichtenthaler, 1992*). However, there has been a lack of synchronous observation for fluorescence kinetic parameters and fluorescence spectrum, which can be used to evaluate the response of leaf to drought stress (*Magney et al., 2017*).

In this study, cucumber was used as an ideal test plant due to its short growth period, easy survival, and because it is widely used in ecophysiology research (*Li et al., 2008*). A drought experiment was conducted over an 8-day from November 24 to December 1, 2018. Gas exchange parameters, fluorescence kinetic parameters, fluorescence spectrum, and chlorophyll content were measured in cumber leaf. Here, our overall objective was to assess the response of photosynthesis traits and ChlF ratio to drought stress based on synchronous observation of gas exchange and fluorescence under drought stress. We hypothesize that (i) photosynthesis would be inhibited by stomatal and non-stomatal limitations under drought stress, (ii) the relationship between the ChlF ratio and chlorophyll content might be changed under drought stress, and (iii) the ChlF ratio can be used to reflect photosynthesis.

## MATERIALS & METHODS

### Plant material and experimental design

Cucumbers (*Cucumis sativus L.*) were used as plant material, which was cultured in plastic seedling pots (12 × 8 × 10 cm) and cultivated in a growth chamber. Growth chamber

temperature was 20–25 °C at day and 15–18 °C at night then light intensity was 1200 µmol m$^{-2}$ s$^{-1}$ with relative humidity (RH) at 75%. The potting soil was a composite culture substrate composed of wood chips, peat, pine bark, and sand. Six mature cucumbers were divided randomly into two treatments: drought and control, with three replicates per treatment. The drought treatment started on November 21, 2018. The soil moisture content ($\theta_g$) of the drought was measured about 8 ± 2% by the weighing method on November 23, 2018, while the soil moisture content of the control was about 15 ± 1%. During the experiment, the plants of the drought treatment were not irrigated, while the plants of the control group were irrigated daily. The upmost, sunlit, dark green, fully unfolded and mainstem leaves were used to measure gas exchange and $F_v/F_m$, and adjacent leaves were used to measure laser-induced chlorophyll fluorescence and Chl$_t$.

## Measurement of the CO$_2$ response curve and chlorophyll fluorescence

Typical A$_n$/C$_i$ curves (light-saturated net CO$_2$ assimilation rate versus intercellular CO$_2$ concentrations) were measured using the Li-6800 portable photosynthesis system (LI-COR Inc., USA) from 8:00 to 11:30 after two days of drought treatment. The upmost fully unfolded, mainstem leaves were measured at leaf temperature of 25 °C, RH of 50–60%, and photosynthetic photon flux density (PPFD) of 1500 µmol m$^{-2}$ s$^{-1}$. The carbon dioxide concentration of the reference chamber was set as 400, 100, 50, 100, 400, 600, 800, 1,000 µmol mol$^{-1}$. A total of 54 A$_n$/C$_i$ curves were taken (i.e., 3 samples per treatment × 7 times per sample × 2 treatments + 6 samples per treatment on the first day × 2 treatments = 54 curves). Before measuring the A$_n$/C$_i$ curve, the leaves were adapted for 5 min at a CO$_2$ concentration of 400 µmol mol$^{-1}$. Measurements of the A$_n$/C$_i$ curve were taken when gas exchange had equilibrated (taken to be when the coefficient of variation for the CO$_2$ partial pressure differential was below 1% between the sample and reference analyzers). This condition was typically achieved within 1–2 min after a stable CO$_2$ concentration had been reached.

$F_v/F_m$ was measured using the Li-6800 fluorescence leaf chamber (LI-COR Inc., USA) connected to an LI-6800 portable photosynthesis system after dark treatment for one night. In the evening before the measurement, the upmost fully unfolded, mainstem leaves used to measure the dark-adapted fluorescence parameters were wrapped with tin foil. The rectangular flash was configured with a red target of 8,000 µmol m$^{-2}$ s$^{-1}$, a duration of 1,000 ms, the output rate of 100 Hz, and a margin of 5 points.

The laser-induced chlorophyll fluorescence system was composed of blue laser light source with a peak emission of 456 nm (Dslaser, China), USB4000 grating spectrometer, VIS-NIR band optical fiber (Ocean Optics, USA), Long-pass optical filter (AT600lp, Chroma Technology Corp, USA), and computer with software (Fig. 1). The light source output power was 40 mW, and the corresponding light source input voltage was 5.7 V. The spectrometer used in the experiment has a resolution of 1.5 nm, an integral time of 3.8 ms–10 s, and detector covers of 200–1,100 nm. The spectrometer was equipped with a USB port on the side, which was connected to the computer and directly powered by the computer. The linear array CCD detector (Toshiba, Japan) of the USB4000 spectrometer has a pixel count of 3648 (*Li, Huang & Zhang, 2009*). SMA905 fiber adapter (DingSuo
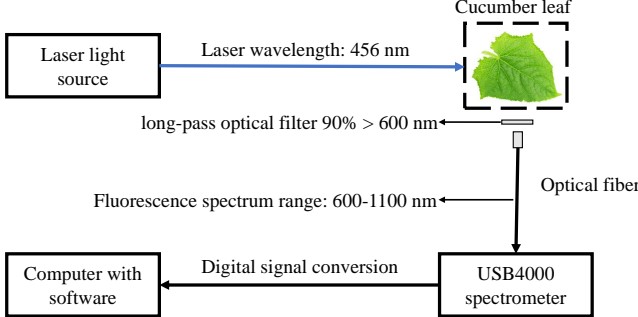

**Figure 1 Schematic diagram of laser-induced chlorophyll fluorescence experimental system.**

Technologies, China) was used as a connector to match the VIS-NIR band optical fiber and USB4000 spectrometer. VIS-NIR band optical fiber has an optical fiber core diameter of 1,000 um, numerical aperture of 0.22, and divergent Angle of 25.4°. The included angle between light source and leaves is 45°, the fiber is perpendicular to the leaves with a distance of 4.5 cm. The long-pass optical filter with a transmission wavelength range greater than 600 nm, and transmittance greater than 90% to prevent the influence of reflected light on the fluorescence spectrum. The chlorophyll fluorescence of 650–850 nm was received by optical fiber then collected by the spectrometer. SpectraWiz software (StellarNet Inc., Tampa FL, USA) was used to set up to collect three spectra and take the average, the integrating time with 600 ms.

## Measurement of chlorophyll content

The leaves used to measure the ChlF spectrum were cut out and used to measure $Chl_t$. Starting from November 24, 2018, $Chl_t$ was measured every other day. The control and drought were repeated three times (repeated six times on the first day), for a total of four measurements. A total of 30 chlorophyll content were collected (i.e., 3 samples per treatment × 3 times per sample × 2 treatments + 6 samples per treatment on the first day × 2 treatments = 30). Acetone and anhydrous ethanol were mixed into the extract at a volume of 1:1. The leaves were cut into filaments and weighed at 0.1 g in the test tube containing the mixture, which was placed in the dark place. After the material was completely white, the optical density at 663 nm and 645 nm was measured by spectrophotometer (MAPADA, China). The $Chl_t$ in this study was expressed as follows:

$$Chl_a = 12.72 \times A_{663} - 2.59 \times A_{645} \times 0.1 \tag{1}$$

$$Chl_b = 22.88 \times A_{663} - 4.67 \times A_{645} \times 0.1 \tag{2}$$

$$Chl_t = Chl_a + Chl_b. \tag{3}$$

Where $Chl_a$ is the chlorophyll A content (mg g$^{-1}$), and $Chl_b$ isthe chlorophyll B content (mg g$^{-1}$), and $Chl_t$ isthe total chlorophyll content (mg g$^{-1}$). The $A_{645}$ and $A_{663}$ are absorbances at wavelengths 645 and 663, respectively.

## Statistical analysis

$V_{cmax}$ and $J_{max}$ were estimated by fitting the $A_n/C_i$ curves using a spreadsheet-based software developed by Sharkey (*Sharkey, 2016*). The chlorophyll fluorescence collection program for the spectrometer was written based on the underlying program of the spectrometer with MATLAB software (*Haiye, Haoyu & Lei, 2009*). Dark current noise is removed from the chlorophyll fluorescence spectrum curve and the curve is smoothed by Savitzky-Golay filtering (*Gorry, 1990*).

We repeated measurements of the same six individuals. Repeated Measures ANOVA (RMANOVA) was used to test the effects of drought stress on photosynthetic traits and chlorophyll fluorescence parameters. The effects were considered to be significantly different if $P < 0.05$. Besides, a mixed-effect linear model was used to evaluate the effect of $V_{cmax}$, $J_{max}$, $g_s$, and $Chl_t$ on $A_n$ and chlorophyll fluorescence parameters. The individual plant was used as a random term. Similar method was used to test the relation between $A_n$ and chlorophyll fluorescence parameters. All statistical analyses were performed using SPSS 25.0 (SPSS Inc., USA).

## RESULTS

### The response of leaf photosynthetic traits and ChlF ratio to drought

The drought stress caused a significant reduction in $A_n$, $V_{cmax}$, $J_{max}$, $g_s$, and $Chl_t$ compared with control ($P < 0.05$). The averages of $A_n$ were $1.0 \pm 0.1$ $\mu$mol m$^{-2}$ s$^{-1}$ and $2.0 \pm 0.2$ $\mu$mol m$^{-2}$ s$^{-1}$ in drought and control (Fig. 2A). The averages of $V_{cmax}$ and $J_{max}$ were $92.0 \pm 5.0$ $\mu$mol m$^{-2}$ s$^{-1}$ and $117.0 \pm 5.0$ $\mu$mol m$^{-2}$ s$^{-1}$ in drought, respectively (Figs. 3A and 3C). For control, the averages of $V_{cmax}$ and $J_{max}$ were $105.9 \pm 4.3$ $\mu$mol m$^{-2}$ s$^{-1}$ and $141.0 \pm 5.2$ $\mu$mol m$^{-2}$ s$^{-1}$, respectively (Figs. 3A and 3C). Compared to control, the averages of $V_{cmax}$ and $J_{max}$ decreased 13.1% and 17.1%, while the $g_s$ and $Chl_t$ in drought ($0.1 \pm 0.01$ mol m$^{-2}$ s$^{-1}$ and $1.6 \pm 0.1$ mg g$^{-1}$) were reduced by 27.1% and 21.5% compared with control ($0.1 \pm 0.01$ mol m$^{-2}$ s$^{-1}$ and $2.1 \pm 0.1$ mg g$^{-1}$) (Figs. 3E and 3G). Compared to the control plants, the averages of $F_v/F_m$ decreased 6.8% ($0.74 \pm 0.01$ to $0.69 \pm 0.13$), while $LD_{685}/LD_{740}$ increased 10.7% ($1.1 \pm 0.4$ to $1.2 \pm 0.3$).

### Control factors for $A_n$ and ChlF ratio

Significant positive correlation was found between $A_n$ and $V_{cmax}$ in the control ($R^2 = 0.15$, $P = 0.03$) and drought stress ($R^2 = 0.45$, $P < 0.001$). There was a significant positive correlation between $J_{max}$ and $A_n$ in the control ($R^2 = 0.12$, $P = 0.04$) and under drought stress ($R^2 = 0.60$, $P < 0.001$). Besides the significant positive correlation ($R^2 = 0.11$, $P = 0.05$) between $g_s$ and $A_n$ in the control, a significant positive correlation ($R^2 = 0.48$, $P < 0.001$) in the drought stress was observed. A poor correlation ($R^2 = 0.07$, $P = 0.84$) between $A_n$ and $Chl_t$ was observed in the control, whereas the correlation was positive ($R^2 = 0.56$, $P < 0.001$) in the drought stress (Figs. 4A–4D).

A significant positive correlation between $F_v/F_m$ and $V_{cmax}$ was found in the drought ($R^2 = 0.13$, $P = 0.03$). Meanwhile, a marginally positive correlation between $F_v/F_m$ and $J_{max}$ was observed in the control ($R^2 = 0.09$, $P = 0.09$), and a significant positive correlation was observed in the drought stress ($R^2 = 0.28$, $P = 0.03$). In addition, a significant positive

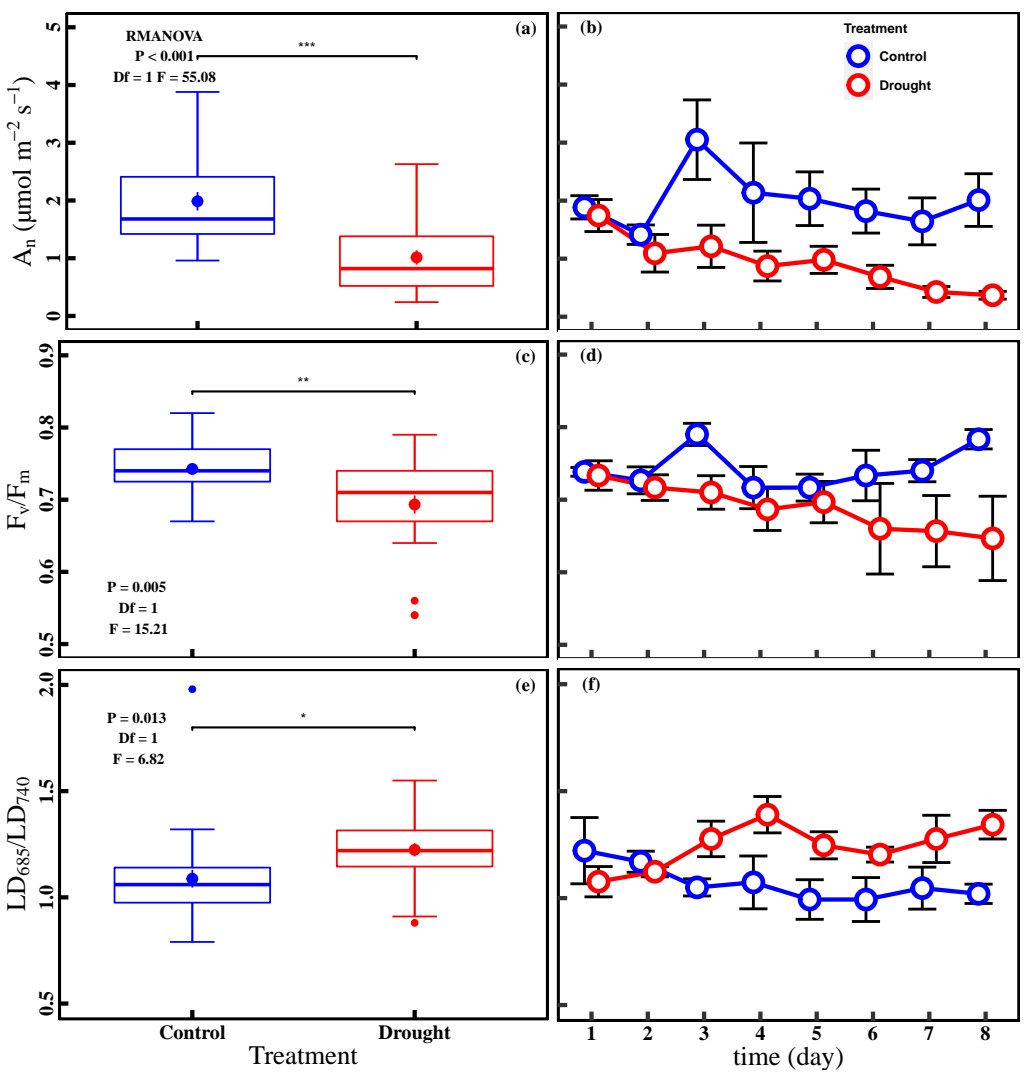

**Figure 2  Variations of net CO₂ assimilation rate and chlorophyll fluorescence parameters in control and drought stress.** (A, B) $A_n$ (net $CO_2$ assimilation rate $\mu$mol m$^{-2}$ s$^{-1}$), (C, D) $F_v/F_m$ (maximum quantum efficiency of PSII), (E, F) $LD_{685}/LD_{740}$ (Laser induced chlorophyll fluorescence intensity ratio). In the box plot, the points and short error bars represent the mean ($\pm$SE) of $n = 27$ per treatment, and the line and long error bars represent the median line and 95% CI, respectively. In the line chart, the points and error bars reflect the mean ($\pm$SE) of three replicates per treatment per date (six replicates per treatment on the first day). The blue and red indicates the control and drought treatment, respectively. RMANOVA was used estimate the effect of treatment : * $P < 0.05$; ** $P < 0.01$; *** $P < 0.001$; n.s. not significant.

correlation was found between $F_v/F_m$ and g$_s$ in the control ($R^2 = 0.28$, $P = 0.003$) while the correlation was poor under drought stress ($R^2 = 0.01$, $P = 0.22$). No significant correlation ($R^2 = 0.03$, $P = 0.27$) was observed between $F_v/F_m$ and Chl$_t$ in the control group, whereas a significant positive correlation was found between $F_v/F_m$ and Chl$_t$ under drought stress ($R^2 = 0.29$, $P = 0.02$) (Figs. 4E–4H).

A significant negative correlation was found between $LD_{685}/LD_{740}$ and V$_{cmax}$ in drought stress ($R^2 = 0.18$, $P = 0.009$), while there was no significant correlation between

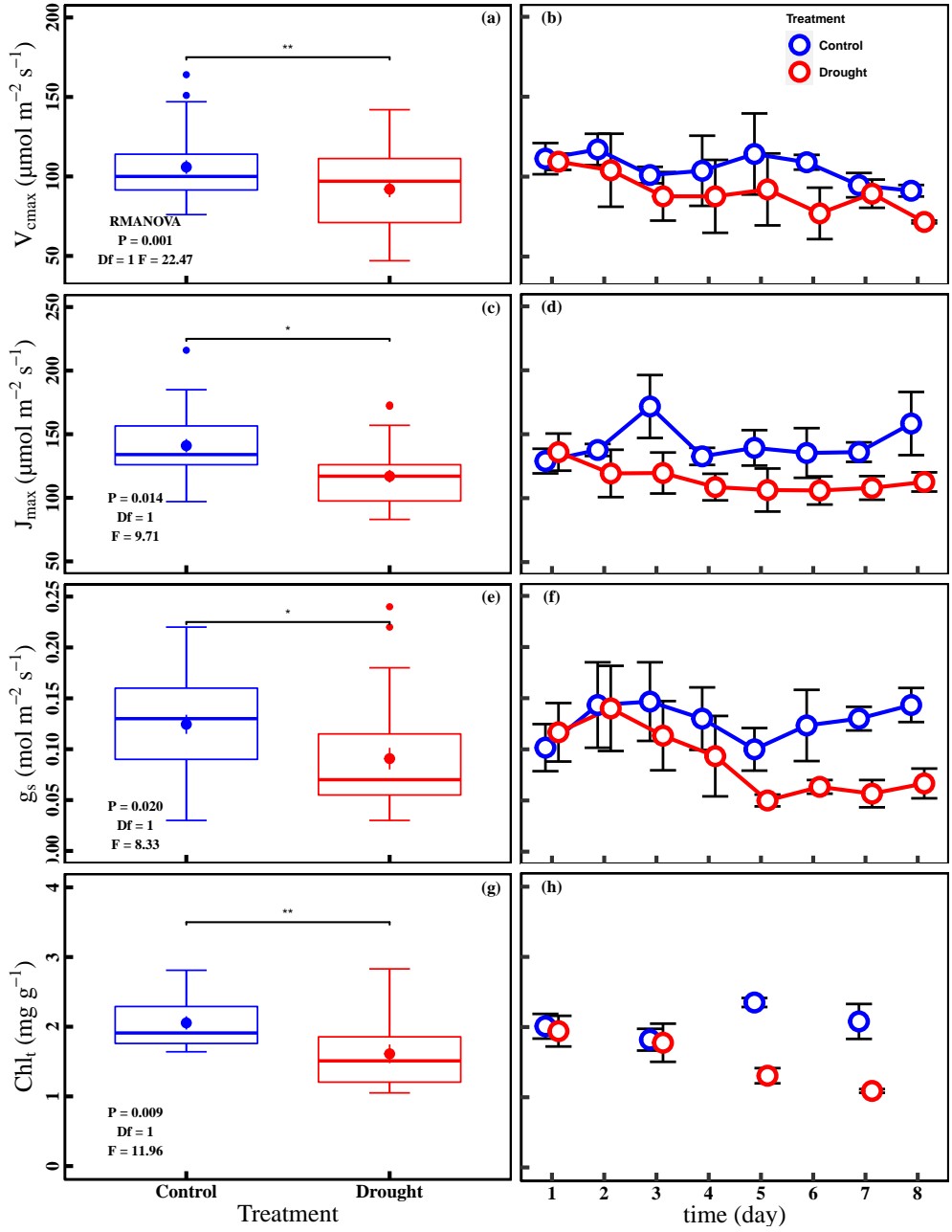

**Figure 3 Variations of photosynthetic traits and total chlorophyll concentration in control and drought stress.** (A, B) Vcmax (Maximum carboxylation rate, $\mu$mol m$^{-2}$ s$^{-1}$), (C, D) Jmax (Maximum photoelectron transfer rate, $\mu$mol m$^{-2}$ s$^{-1}$), (E, F) gs (stomatal conductance, mol m$^{-2}$ s$^{-1}$), (G, H) Chlt (Total chlorophyll concentration, mg g$^{-1}$) in control and drought stress. In the box plot, the points and short error bars represent the mean ($\pm$SE) of $n = 27$ per treatment ($n = 15$ of Chl$_t$ per treatment), and the line and long error bars represent the median line and 95% CI, respectively. In the line chart, the points and error bars reflect the mean ($\pm$SE) of three replicates per treatment per date (six replicates per treatment on the first day). The blue and red indicate the control and drought stress, respectively. RMANOVA was used estimate the effect of treatment : * $P < 0.05$; ** $P < 0.01$; *** $P < 0.001$; n.s. not significant.

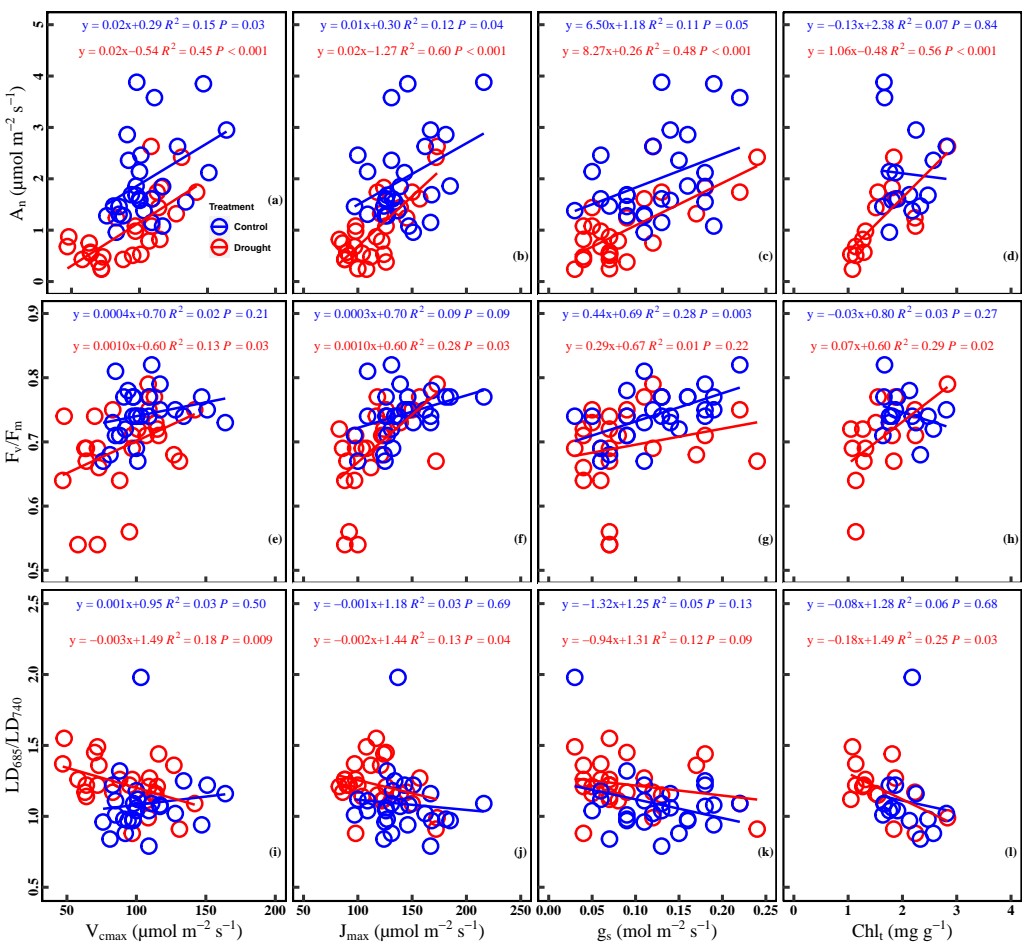

**Figure 4** **Relationships between chlorophyll fluorescence parameters and photosynthetic traits in the control and drought stress.** (A) $A_n$ (net $CO_2$ assimilation rate $\mu$mol m$^{-2}$ s$^{-1}$) and $V_{cmax}$ (Maximum carboxylation rate, $\mu$mol m$^{-2}$ s$^{-1}$), (B) $A_n$ and $J_{max}$ (Maximum photoelectron transfer rate, $\mu$mol m$^{-2}$ s$^{-1}$), (C) $A_n$ and $g_s$ (stomatal conductance, mol m$^{-2}$ s$^{-1}$), (D) $A_n$ and $Chl_t$ (Total chlorophyll concentration, mg g$^{-1}$), (E) $F_v/F_m$ (maximum quantum efficiency of PSII) and $V_{cmax}$, (F) $F_v/F_m$ and $J_{max}$, (G) $F_v/F_m$ and $g_s$, (H) $F_v/F_m$ and $Chl_t$, (I) $LD_{685}/LD_{740}$ (Laser-induced chlorophyll fluorescence intensity ratio) and $V_{cmax}$, (J) $F_v/F_m$ and $J_{max}$, (K) $F_v/F_m$ and $g_s$, (L) $F_v/F_m$ and $Chl_t$. Linear fitting was used for correlation analysis ($n = 27$ for per treatment and $n = 15$ of $Chl_t$ for per treatment). The blue line and red line indicate the linear regression for the control and drought stress, respectively.

$LD_{685}/LD_{740}$ and $V_{cmax}$ in control ($R^2 = 0.03$, $P = 0.50$). There was no significant correlation between $LD_{685}/LD_{740}$ and $J_{max}$ in control ($R^2 = 0.03$, $P = 0.69$), while a significant negative correlation was observed between $LD_{685}/LD_{740}$ and $J_{max}$ in drought stress ($R^2 = 0.13$, $P = 0.04$). No significant correlation was observed between $LD_{685}/LD_{740}$ and $g_s$ in control ($P = 0.13$) and drought stress ($P = 0.09$). There was no significant correlation between the $LD_{685}/LD_{740}$ and $Chl_t$ in the control ($R^2 = 0.06$, $P = 0.68$), while a significant negative correlation between $LD_{685}/LD_{740}$ and $Chl_t$ in drought stress ($R^2 = 0.25$, $P = 0.03$) (Figs. 4I–4L).

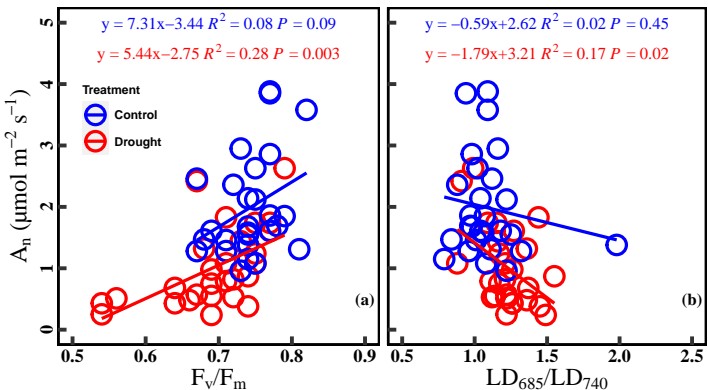

**Figure 5** **Relationships between net CO₂ assimilation rate and chlorophyll fluorescence parameters in control and drought stress.** (A) $A_n$ (net $CO_2$ assimilation rate $\mu$mol m$^{-2}$ s$^{-1}$) and $F_v/F_m$ (maximum quantum efficiency of PSII), (B) or $A_n$ and $LD_{685}/LD_{740}$ (Laser-induced chlorophyll fluorescence intensity ratio). Linear fitting was used for correlation analysis ($n = 27$ for per treatment). The blue line and red line indicate the linear regression for the control and drought stress, respectively.

## Correlation between $A_n$ and ChlF ratio

There was a marginally positive correlation between $A_n$ and $F_v/F_m$ in the control ($R^2 = 0.08$, $P = 0.09$). However, the correlation between $A_n$ and $F_v/F_m$ was significant positive in drought stress ($R^2 = 0.28$, $P = 0.003$). Similarly, there was no significant correlation between $A_n$ and $LD_{685}/LD_{740}$ in the control ($R^2 = 0.02$, $P = 0.45$), while a significant negative correlation was found between $A_n$ and $LD_{685}/LD_{740}$ in drought stress ($R^2 = 0.17$, $P = 0.02$) (Figs. 5A–5B).

## DISCUSSION

### Combination of the stomatal limitation, non-stomatal limitations, and chlorophyll content regulated leaf photosynthesis under drought stress

As expected, drought stress significantly decreased the leaf photosynthesis of cucumber (Fig. 2A). Although drought stress is known to reduce leaf photosynthesis, the processes responsible for the key limitations are still a matter of debate (*Chaves et al., 2002*; *Flexas & Medrano, 2002*; *Pinheiro & Chaves, 2011*). Here, our study found that the combination of the stomatal limitation, non-stomatal limitations, and chlorophyll content regulated the decrease of leaf photosynthesis under drought stress (Figs. 4A–4D). Increasing evidence shows that leaf photosynthesis under drought stress is not limited by a single process (*Zhou et al., 2015*). It has been demonstrated that stomatal closure reduces photosynthesis and transpiration while improving water use efficiency due to acclimation under drought stress (*Feller, 2016*; *Lamaoui et al., 2018*). Non-stomatal limitations were defined as the sum of the contributions of mesophyll conductance and leaf biochemistry, which directly reflect the biochemical process of photosynthesis (*Grassi & Magnani, 2005*). The decrease in $V_{cmax}$ and $J_{max}$ may result from a decrease in the amount of active Rubisco and an inadequate supply of ATP or NADPH or to a low enzymatic activity during the photosynthetic carbon

reduction cycle (*Campos et al., 2014*; *Flexas et al., 2004*; *Peña Rojas, Aranda & Fleck, 2004*). Recent studies found that stomatal and non-stomatal limitations to photosynthesis are coordinated on similar timescales, and suggested that non-stomatal limitations should be included in predict the model of photosynthesis response to drought (*Drake et al., 2017*; *Salmon et al., 2020*). A study of three Mediterranean species has found that the decrease of $A_n$ was simultaneous regulated by stomatal and biochemical limitations during drought stress (*Varone et al., 2012*). In addition, $Chl_t$ is the main pigment that absorbs photosynthetically active radiation and can indirectly reflect the integrity of the photosynthetic device (*Streit et al., 2005*). $Chl_t$ can be used as a functional trait to evaluate drought stress (*Pilon et al., 2018*; *Pilon et al., 2014*). Therefore, our observations suggest that the combination of the stomatal limitation, non-stomatal limitation, and chlorophyll content should be taken into consideration in process-based models for simulating photosynthesis in terrestrial ecosystems under drought stress.

### $J_{max}$ and $Chl_t$ governed ChlF ratio under drought stress

Leaf $F_v/F_m$ is a crucial chlorophyll fluorescence parameter for evaluating the health or integrity of the internal apparatus under drought stress (*Krause & Weis, 1991*; *Urban, Aarrouf & Bidel, 2017*). Here, we found that $F_v/F_m$ was significantly decreased under drought stress (Fig. 2B), which revealed that the PSII may be damaged under drought stress, and the primary reaction of photosynthesis may be inhibited (*Lichtenthaler & Rinderle, 1988*). The fluorescence parameters of the leaves are changed in two ways under stress conditions. The minimal fluorescence ($F_o$) increases due to obstruction of the electron flow through PSII, and plastoquinone acceptor ($QA^-$) cannot be completely oxidized during stress. Simultaneously, the reduction of $F_m$ during stress may be affected by decreased activity of the water-splitting enzyme complex and perhaps a concomitant cyclic electron transport within or around PSII (*Porcar-Castell et al., 2014*). Therefore, $F_v/F_m$ will decrease under drought stress. Our finding was consistent with a previous studying, in which drought stress inhibited the photochemical activity of PSII and decreased leaf $F_v/F_m$ (*Meng et al., 2016*). Meanwhile, our study showed that $F_v/F_m$ was largely related to $J_{max}$ and $Chl_t$ under drought stress (Figs. 4F and 4H). It has been proposed that $J_{max}$ is decreased by drought stress, which prevents the electron from rapidly transferring back, and hinders the whole photochemical process (*Baker & Rosenqvist, 2004*; *Khatri & Rathore, 2019*). Similarly, the decrease in $Chl_t$ will weaken the photochemical process, which demonstrates the dependence of the light absorption and fluorescence emission on the concentration of chlorophyll molecules in the chloroplast (*Nyachiro et al., 2001*). Thus, the significant linear relationship between $F_v/F_m$ and $J_{max}$ and $Chl_t$ observed in drought stress jointly indicates the importance of $J_{max}$ and $Chl_t$ in governing chlorophyll fluorescence.

In our study, the $LD_{685}/LD_{740}$ based on spectral analysis was significantly increased under drought stress (Fig. 2C). This finding was similar to the study of *Meng et al. (2016)*, in which the fluorescence intensity ratio increased when the PSII was damaged. Moreover, $LD_{685}/LD_{740}$ was largely regulated by $J_{max}$ and $Chl_t$ under drought stress (Figs. 4J and 4L). The previous study found that changes in the chlorophyll content resulted in changes of more than 90% for the $F_{690}/F_{735}$ ratio (*Csintalan, Tuba & Lichtenthaler, 1998*). There was

a significant negative correlation between the $LD_{685}/LD_{740}$ and $Chl_t$ under drought stress (Fig. 4L). The chlorophyll absorption spectrum overlaps with the chlorophyll fluorescence emission spectrum in the red band, which results in the $LD_{685}$ being decreased by re-absorption in the case of higher chlorophyll content. The effect of re-absorption on the red band is stronger than that of the far-red band, and therefore the $LD_{685}/LD_{740}$ will decrease (*Buschmann, 2007*). The $LD_{685}/LD_{740}$ represents an ideal tool for evaluating the change of $Chl_t$ and reflects the photochemical activity of PSII indirectly under drought stress (Figs. 4J and 4L). Spectral analysis has been used to directly assess ecosystem functioning under climate change. For instance, *Gameiro et al. (2016)* found a significant linear relationship between the fluorescence intensity ratio and leaf water content of *Arabidopsis*. *Norikane et al. (2003)* successfully monitor the growth of tomato under drought stress based on spectral analysis. Here, synchronous observation of $LD_{685}/LD_{740}$ and $F_v/F_m$ based on spectral analysis and fluorescence kinetics suggest that $LD_{685}/LD_{740}$ can be used as an indicator for detection of plant stress.

## Drought stress strengthens the relationship between net $CO_2$ assimilation rate ($A_n$) and ChlF ratio

Our study reported a significant relationship between the $A_n$ and ChlF ratio under drought stress, while no significant correlation was found in the control (Figs. 5A and 5B). The strengthening relationship between $A_n$ and ChlF ratio may be ascribed to variations of $J_{max}$ and $Chl_t$ under drought stress. On the one hand, the reduction of $F_v/F_m$ and $J_{max}$ under drought indicated that the photosynthetic electron transport was damaged. Drought stress damages the reaction center of PSII and inhibits the electron transfer process of photosynthesis, which reduces the light energy conversion efficiency of PSII (*Brestic et al., 1995*; *Cornic & Fresneau, 2002*; *Longenberger et al., 2009*). On the other hand, drought stress deforms the leaf chloroplast layer structure and reduces chlorophyll content (*Batra, Sharma & Kumari, 2014*). So, $J_{max}$ and $Chl_t$ became limiting factors for the $A_n$ and ChlF ratio under drought stress (Figs. 4B, 4D, 4F, 4H, 4J and 4L). The strengthening relationship between $A_n$ and ChlF ratio has been observed in previous studies (*Murchie & Lawson, 2013*; *Su et al., 2015*). For example, *Wang et al. (2018)* found a significant linear relationship between $A_n$ and $F_v/F_m$ in the soybean experiment under drought conditions. *Batra, Sharma & Kumari (2014)* found similar results by studying the ChlF ratio characteristics of mung beans under drought conditions. Therefore, ChlF ratio based on spectral analysis and fluorescence kinetics was a better indicator of the photosynthetic capacity under drought stress.

## CONCLUSIONS

Our results demonstrate that the decrease in cucumber leaf photosynthesis is regulated by stomatal limitation, non-stomatal limitation, and chlorophyll content under drought stress. We recommend incorporating the effects of stomatal, non-stomatal limitations and chlorophyll content, and applying them to the prediction of plant photosynthesis response to drought stress. The $J_{max}$ and $Chl_t$ are key limiting factors for the ChlF ratio under drought stress, and the ChlF can characterize plant photosynthetic capacity as new technology under drought stress.

## ACKNOWLEDGEMENTS

We gratefully acknowledge Yunhui Tan, Meijuan Fu, and Yao Si for their help with the measurement of the photosynthesis and chlorophyll fluorescence. We thank two anonymous reviewers and editor for their constructive comments.

### Funding

This study was supported financially by the Zhejiang Provincial Natural Science Foundation of China (LY19C030004), the National Key Research and Development Program of China (2017YFB0504000 and 2016YFB0501501), the National Natural Science Foundation of China (41871084 and 31400393), and a grant from State Key Laboratory of Resources and Environmental Information System. The funders had no role in study design, data collection and analysis, decision to publish, or preparation of the manuscript.

### Grant Disclosures

The following grant information was disclosed by the authors:
Zhejiang Provincial Natural Science Foundation of China: LY19C030004.
National Key Research and Development Program of China: 2017YFB0504000, 2016YFB0501501.
National Natural Science Foundation of China: 41871084, 31400393.
State Key Laboratory of Resources and Environmental Information System.

### Competing Interests

The authors declare there are no competing interests.

### Author Contributions

- Jie Zhuang and Yonglin Wang conceived and designed the experiments, performed the experiments, analyzed the data, prepared figures and/or tables, authored or reviewed drafts of the paper, and approved the final draft.
- Yonggang Chi and Lei Zhou conceived and designed the experiments, performed the experiments, analyzed the data, authored or reviewed drafts of the paper, and approved the final draft.
- Jijing Chen, Wen Zhou, Jun Song, Ning Zhao and Jianxi Ding analyzed the data, prepared figures and/or tables, and approved the final draft.

### Data Availability

The raw measurements are available in Supplemental File.

### Supplemental Information

Supplemental information for this article can be found online at http://dx.doi.org/10.7717/peerj.10046#supplemental-information.

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
