# Peer review of "Drought stress strengthens the link between chlorophyll fluorescence parameters and photosynthetic traits"

_PeerJ, doi:10.7717/peerj.10046_

## Round 0.1 · original submission · Major Revisions

I have now received two detailled reviews for your manuscript. Both reviewers find value in your paper and particulalry in your dataset but also highlight the need for extensive revisions of the manuscript.
In addition to the multiple points raised by the reviewers, one of my main concern is the lack of a clearly scientifically motivated questions, as pointed out by reviewer 2, there are multiple papers on the topics. Confirmative studies are welcome in PeerJ but they still need a clear objective and address important scientific questions. Clarifying your objectives and possibly making hypotheses in the introduction should help improve the message of the manuscript. This lack of a clear hypothesis also impacts the discussion which at the moment is of limited interest. I also agree with the impression mentioned by reviewer 1, that the manuscript seems to have been written in a rush.

Therefore, substantial improvements are needed before the manuscript can be reassessed and my decision is major revision required.

Should you decide to revise and resubmit your manuscript, I would encourage you to prepare a clear and complete manuscript, by providing enough details to allow the study and data analyses to be replicated.



Some additional specific comments:

introduction:
- not very well written the logic flow is not clear.
L58-60, yes but there has been recent publication on the topic: e.g., Drake et al. Agric Forest Met 2018, Salmon et al. 2020 New Phytologist (apologies for self-citation here, but the authors might find it relevant. Do not feel that you need to cite it).
L81 cite some examples.
L82 ecophysiology: one word
L85-87 it is not very clear from the introduction why those objective.

Material and methods:
too little details to allow repetition: for example numbre of replicates, repeated measurments on same plants ot not, etc. This should be the guideline of describing your work here.

Results
Given that your objective is to quantitatively analyse the response, why not report present/discuss the quantitative response/relations and only R2 and p?
what is interval values provided after the mean?
L157decreased
the relatively poor correlation between A and gs is surprising. Any explanation?
More than just R2 and p-values, it would be interesting to know how are the relation quantitatively (in particular the slope)
Discussions
L203 they are mode recent reference (see for example above, with some elements of response)
L206 how do you define non stomatal limitation? How changes in chlorophyll concentration difers from it?

Looking at figure 2, A result suggest a rewatering of the control rather than a stress

Conclusion
L263 see Gimeno et al. 2019 Journal Exp. Bot.

Reviewer 1 ·

Basic reporting

Some text need to be improved to make sure it could be easily understood. Some literature are not cited appropriately.

Experimental design

How the research question can fill the knowledge gap is not well introduced. The experiment is well designed, but missing sufficient detail.

Validity of the findings

The results and conclusions are not well stated, but could be improved after revising.

Additional comments

Apparently, the authors wrote this manuscript very quick. The experiment is well designed, but some important information of the measurement are missing. You did have good results, but the results and discussion are not well written. The main comment is the authors should enhance their understanding about chlorophyll fluorescence, the title, results and discussion should be carefully revised. I would expect this manuscript can be improved by next version.

Introduction
L61: you did find some literature showed that drought had an effect on Fv/Fm, but what the mechanism behind it and how drought will/may affect chlorophyll fluorescence are not well addressed.
L77: why fluorescence ratio can reflect the photosynthetic activity?
L78: why they were negatively correlated with each other?

Materials and methods
L91: please add ‘Latin name’ of the plant species.
L96: How the leaf samples are selected, size, age (light green or dark green leaf), leaf position in the plant (e.g. top, middle or bottom), sunlit or shade, etc? All these factors will significantly affect the physiological variations, and fully unfold also does not mean mature leaves. Are same or different leaves used to conduct gas exchange and laser fluorescence measurement?
You had eight days measurements (I know it when I saw your figures), but this is not mentioned in the Materials and methods section. Are these days continuously or not? Did the plants keep the same conditions during the eight days experiment for each treatment or the drought level was enhanced day by day. What is the sample size? How many seedlings you had for each treatment? How many replicates you had for each measurement (e.g. gas exchange, chlorophyll content, etc.)?
L111: please add the company name and country of the instrument (LI-6800 PAM fluorometer) to measure Fv/Fm.
L118: which laser system (e.g. instrument, company, etc.)? which light source? What pulse and at which wavelength you used to induce fluorescence? Which optical filter you used for excluding the radiation to get fluorescence spectra in which range wavelength?
L121: what exactly integrating time you used for your experiment? What do you mean ‘optimal operating range’? how many spectra you sampled to estimate fluorescence?
L124: please clarify what is ‘SMA905 interface’?

Results
L151: you showed how ChlF ratio rather than ChlF in response to the drought.
L152: what the values (e.g., 1.01±0.12 μmol m-2 s-1) mean? I am confused what P value indicated exactly, do you mean An had a significant decrease in drought treatment across your eight days experiments, or during the whole experiment period An in drought treatment was significant higher than control. Apparently An was significant decreased in drought treatment in your eight days experiments. Please clarify all these (values of each variable in drought and control, and p values) for all the variables in these two paragraphs until line 163.
At day 3, a very high An was observed, together with a relatively high Fv/Fm. This is due to they were measured at the same leaf using LICOR. But corresponding changes in fluorescence ratio and total Chl were not found. Can you explain why a high An and Fv/Fm at day 3 were observed and also why others not (this can be discussed in discussion or describe in Materials and methods).
L190: you did not compare the correlations between photosynthesis and ChlF, but net photosynthesis rate with maximum photochemical efficiency and fluorescence ratio. If you want to analysis ChlF, you should show the results of single band ChlF (e.g. ChlF at 685 nm or 740).

Discussion
Section 1: the discussion is very vague. Yes, under drought stress, stomatal, non-stomatal and Chl regulated the changes of photosynthesis, but why and how they contribute to photosynthetic activity are not discussed.
Section 2: the title of the section is not correct because the first paragraph is about Fv/Fm which is maximum photochemical efficiency or photosynthetic capacity, not fluorescence. In line 222, decrease in total Chl will weaken the photochemical activity, but is it mainly due to Chl contribute to the emission of Chl fluorescence? How you know PSII fluorescence contribute 95% of total Chl fluorescence emission (carefully cite the literature), note that how PSI and PSII contribute to the total fluorescence emission are very complicate and still not clearly understood now.
L227: why fluorescence ratio can evaluate the photosynthetic activity? I strongly recommend the authors to enhance the understanding what is the fluorescence and how and why fluorescence or parameters calculated from fluorescence can be used to indicate the photosynthesis, not only how good relationship found by other studies or how the stress changed the parameters, the reasons behind that are most important and should be discussed. For example, the decrease of ChlF ratio in the drought treatment probably related to the decrease of Chl concentration, because less Chl means less emission of ChlF both in red and far-red band and at the same time less reabsorption of fluorescence in the red band, namely the relatively decrease level of ChlF emission in red band will be less than that in far-red band, leading to an increase of fluorescence ratio. You can find many literature showed that ChlF ratio could be a good indicator of chlorophyll content.
Section 3: again, you did not analysis the relationship between photosynthesis and ChlF. Please check this throughout the manuscript.

Figures:
Figure 1: are you sure this is a cucumber leaf? This diagram is really rough. I cannot see how you use laser induce the fluorescence and how this diagram can help the reader to easily understand the experiment design.
Figures 2-3: please clarify what P value indicates in the figure caption, (the difference between the two treatments?). Please also add what error bar indicates (standard error or standard deviation) and how many replicates were used to calculate the error bar
Figure 2b and L154: Again, which stage and condition of the leaf was selected as sample? From the changes of Fv/Fm, it seems that the sample was still not mature or had some stress at the beginning of the experiment. Ideally, Fv/Fm values for a health and mature leaf should at least 0.75. If the value during the peak growing season is less than 0.75, which means the plant may suffer from certain stress or photoinhibition, etc. Your Fv/Fm are mostly less than 0.75 in both two treatments. Thus, I doubt how you selected your samples, which should also be clarified in the Materials and methods section.
Figure 3d: Chlorophyll is an important factor to control the chlorophyll fluorescence and photosynthetic activity, but why you only had four measurements? This should be described either here or in Materials and Methods section.
Figure 4: Please clarify each result of linear regression model belong to which treatment, either by different colors or adding descripting in the figure caption.

References: please carefully check the citation. For example, in line 62, Sperdouli & Moustakas 2012 is not the one should be cited here, and in lines 70-72 you talked results from Li et al., 2008 but you cited another reference ‘Tribulato et al. 2019’ here.

Here are some recommended literature:
Buschmann 2007. Variability and application of the chlorophyll fluorescence emission ratio red/far-red of leaves. https://link.springer.com/article/10.1007/s11120-007-9187-8
Lichtenthaler & Rinderle. The Role of Chlorophyll Fluorescence in The Detection of Stress Conditions in Plants. https://www.tandfonline.com/doi/abs/10.1080/15476510.1988.10401466
Magney et al., 2017. Connecting active to passive fluorescence with photosynthesis: a method for evaluating remote sensing measurements of Chl fluorescence. https://nph.onlinelibrary.wiley.com/doi/full/10.1111/nph.14662

Reviewer 2 ·

Basic reporting

Many similar papers using chlorophyll fluorescence as an indicator of water stress in plants were already published ( doi.org/10.1007/BF00203643, doi: 10.3389/fphys.2019.00786, doi.org/10.1093/aob/mcf064, DOI: 10.1093/jexbot/51.345.659 ).
The structure of the paper is good and acceptable. The amount and quality of the results presented and the style of the figures are very good.
The novelty of the paper is confusing, not very clear, however, the paper is interesting.

Experimental design

The main aim of the paper was to introduce the relationships among the parameters of chlorophyll fluorescence and photosynthesis. The hypothesis is not very new, but the scientific problem is still actual.
Authors realized the pot experiments with cucumber, however, conclusions are hyperstimated.

The authors studied the effect of drought on photosynthetic parameters. They should write exactly, how they controlled/measured leaf water status.

Validity of the findings

It is needed to add information about number of repetitions of the measurements and experiments.
Overall, the results are presented in a comprehensible manner, with supporting charts and graphs to illustrate the statistical analyses. In Discussion, the obtained results have not been sufficiently compared with analogical researches.

Additional comments

It has often been published that the maximum quantum yield of PSII photochemistry measured by the ratio Fv/Fm is not changed even by a severe drought. The decline of the photosynthesis during dehydration is attributed to stomatal closure. Dehydration of the leaves during progressive drought up to 1,5 MPa had no effect on the quantum yield of CO2-dependent O2 evolution.
The gradual reduction of photochemical quenching (qP) and quantum efficiency (ΦPS2) was observed under drought stress while non-photochemical quenching increased. The maximum efficiency of PSII (Fv/Fm) was not affected by drought stress (doi.org/10.1007/s11099-008-0108-7).
On the other hand, chlorophyll fluorescence researchers use for the signalization of changes in photosynthetic apparatus under different harmful environmental situations (doi.org/10.3390/w12010289, DOI: 10.32615/ps.2019.179, DOI: 10.32615/ps.2020.014 ).
The paper should include more relevant references discussing characterizing changes in the physiological parameters of leaves, photosynthesis, and photochemistry derived from chlorophyll fluorescence measurements.

This study is well designed, and the data are presented in good quality. The understanding of mechanisms is limited, as it is restricted to papers that have a particular view and deliberately ignore alternatives and does not present a balanced view of the evidence. You have to explain more about the mechanisms of plant sensitivity on water deficit.
The discussion lacks a bit in-depth.
I would have expected a more critical discussion of the results.
strengths: good philosophy and analytical work
limitations: lack of novelty

---

## Round 0.2 · Minor Revisions

Both reviewers and I have noted the improvement you made to your manuscript. Reviewer 1 had a number of small comments that I would like you to address. Furthermore, I agree with reviewer 1 that the language should be carefully checked (ideally by a native-equivalent speaker or a professional translator). This is important as it makes it difficult to understand some parts of your text (see comments below). I look forward to reading a revised version of your manuscript.

A few additional small comments:

Pease pay attention to the logic/wording, the text gets somewhat confusing or unclear in places. Here is an example with the first paragraph, with my points in brackets. Please pay attention and apply the same critical thinking in the whole manuscript. I suspect this might be a language issue and here again help from a native-equivalent speaker or professional translator might solve a lot of those issues.

“Drought stress is expected to continue [mostly drought stress is not continuous, the duration and frequency of drought are expected to increase] due to global warming (Pachauri et al. 2014). Drought stress limits leaf photosynthesis and plant productivity (Akhkha et al. 2011; Tezara et al. 2003), increases the frequency of forest fires and the death rate of trees (Anderegg et al. 2013; Phillips et al. 2009), and damages the balance of carbon budget [what does damage the balance of Carbon budget mean? Do you mean something like it will decrease the GPP/NEP?] in terrestrial ecosystems (Buermann et al. 2014; Fang et al. 2015). The terrestrial ecosystem productivity is driven by leaf photosynthesis and is linked [what is the link?] to chlorophyll fluorescence (ChlF) (He et al. 2019; Murchie & Lawson 2013). ChlF was used to extend the interpretation of leaf photosynthetic traits [how?], and then the productivity of canopy or ecosystems is inferred [how?] (Aasen et al. 2019; Smith et al. 2018; Yang et al. 2017). Therefore, better understanding the link between ChlF ratio and photosynthetic traits under drought stress is crucial for accurately projecting the impacts of climate change on terrestrial ecosystems. “

L51 “On the one hand, drought stress affects the diffusion process of CO2 then reduces An through stomatal closure” diffusion through the mesophyll also limits An

L57 scientific names should be italicized

L97-100 merge both sentences

L105 what is the logical link between these sentences?

L112 fine, but what does this mean for the main idea of your paragraph? Please remember that in scientific writing we try to have one idea per paragraph. I think that in dealing with the reviewer comments, you have place different bit of information together without necessarily keeping the logic of the section. Maybe by rearranging (e.g. background information of process on one side, and their usage on the other) the flow would improve.

L119 ideal plant doesn’t mean much. Ideal for what? “ideal test plant”?

L121 “ ecophysiology physiological” pick one

L209 reference for MATLAB is missing

L212-216 if I understood correctly your sampling design, you have repeated measurements on the same six individuals. If that’s true, you need to account for it in your analyses, as it violates the assumptions on the ANOVA and linear regressions.

L313-321 this is general discussion, please link to your results.

Reviewer 1 ·

Basic reporting

This version of the manuscript has been improved a lot. The English should be carefully checked throughout.

Experimental design

It is well described now.

Validity of the findings

The results and conclusions have been improved after revising.

Additional comments

I only have few comments for the author.

(1). Please carefully check the English grammar for the whole manuscript.
For example, change ‘than’ to ‘that’ in L104, remove second ‘to’ in L117, change ‘hypothesized’ to ‘hypothesize’ and ‘Photosynthesis’ to ‘photosynthesis’ in L118.
(2). L100-102: in L100 you described ‘The decrease of Chlt will increase the ChlF under stress conditions’, whereas in L315 it was ‘LD740 increases with increasing chlorophyll concentration’. Please clarify these converse logic.
The logic of the following sentence is also not correct. “the ratio of ChlF will increase, this is because of the reabsorption by far-red region of chlorophyll of radiation emitted by red region (Baker 2008; Buschmann 2007)”.
(3). L116: this study was based on leaf level that should be clearly indicated both in Abstract and objectives.
(4). L173: the chlorophyll a fluorescence are mainly emitted between 650 and 850 nm.
(5). L313-315: ChlF at red band can decrease or slightly increase (but the increased level is relatively smaller than at far-red band) due to the increased reabsorption of red ChlF as the increase of the chlorophyll content. But I am not sure if it will be stabilized.

Reviewer 2 ·

Basic reporting

The introduction provides a good understanding of the subject and its importance, with a significant quantity of information.
The article is presented in an intelligible fashion and is written in standard English partially.
The authors used good references and sufficient background for the experimental work. The literary structure is good.
Conclusions are presented in an appropriate fashion and are supported by the data.

Experimental design

The experimental methodology was appropriate and scientific, and the analyses were done correctly. The authors used very progressive techniques and protocols.
The results of the experiments provide novel data and the results and discussion section is deep and complex.
In Discussion, the obtained results have been sufficiently compared with analogical researches.

Validity of the findings

Overall, the study is of good quality and the results are innovative.
Experiments, statistics, and other analyses are performed to a high technical standard and are described in sufficient detail.

Additional comments

I checked the status, reviews, authors' comments on this paper.
The authors have made a great effort in introducing amendments to the manuscript in response to all the reviews. The quality of the manuscript has been significantly improved.
Results reported have not been published elsewhere.
Experiments, statistics, and other analyses are performed to a high technical standard and are described in sufficient detail.
Conclusions are presented in an appropriate fashion and are supported by the data.
The article is presented in an intelligible fashion and is written in standard English partially.
I am satisfied with the revised version, which in my opinion is ready for publication.

---

## Round 0.3 · Minor Revisions

Dear authors, the manuscript has clearly improved since the previous version and the reviewers concerns as well as most of mine have been addressed. Except for a few language points (See below), my only issue is with the statistical analyses. The repeated ANOVA is an improvement, but the linear regression should also account for the repeated measurements. A mixed-effect model with the individual plant as random term would be the way to go.

Small points:

check the number of decimal that are relevant to give considering the precision of the measurements etc. I suspect that one is enough for gas exchange values for example. Same goes for the percent values etc.

Make sure that the figures are easy to read when scaled for publication.

L22 is adopted the right word?

L36 “under drought stress” would better reflects your results

L45 regional scale

L56 space missing before (gm)

L61 delete “the” before gm

L73 and is then consumed

L86 Low Fv/Fm represents situations in which large amount (?)

L161 product reference for the CCD

L243 negative R2 is impossible

L286 a previous studying

L287 there is correlation, but that doesn’t prove the causality. I would avoid “controlled”.

---

## Round 0.4 · accepted · Accept

Thanks for the last revisions. Your manuscript is now ready for publication.